# Research Implications for Future Telemedicine Studies and Innovations in Diabetes and Hypertension—A Mixed Methods Study

**DOI:** 10.3390/nu12051340

**Published:** 2020-05-08

**Authors:** Patrick Timpel, Lorenz Harst

**Affiliations:** 1Department for Prevention and Care of Diabetes, Faculty of Medicine Carl Gustav Carus, Technische Universität Dresden, 01307 Dresden, Germany; 2Research Association Public Health Saxony/Center for Evidence-Based Healthcare, Faculty of Medicine Carl Gustav Carus, Technische Universität Dresden, 01307 Dresden, Germany

**Keywords:** telemedicine, diabetes, hypertension, research needs, mixed method, qualitative content analysis, factor analysis, joint display

## Abstract

(1) Background: The objective of this study was to identify, categorize and prioritize current implications for future research in the use telemedicine for diabetes and hypertension in order to inform policy and practice decisions. (2) Methods: An iterative mixed methods design was followed, including three consecutive steps: An updated umbrella review of telemedicine effectiveness, qualitative content analysis of extracted data on current research needs and a quantitative survey with practitioners and health care researchers in order to prioritize the identified needs. (3) Results: Overall, 32 included records reported on future research implications. Qualitative content analysis yielded five categories as well as subcategories, covering a need for high quality studies, comprehensive technology assessments, in-depth considerations of patients’ characteristics, ethics and safety as well as implementation strategies. The online survey revealed that the most pressing future research needs are data security, patient safety, patient satisfaction, implementation strategies and longer follow-ups. Chi² statistics and *t*-tests revealed significant differences in the priorities of participants with and without experience in telemedicine use, evaluation and development. A factor analysis revealed six over-arching factors. (4) Conclusion: These results may help learning from mistakes previously made and may serve as key topics of a future telemedicine research agenda.

## 1. Introduction

Telemedicine, as one variety of digital health, is characterized by three criteria: (1) using information and communications technology (ICT), (2) covering a geographical distance and (3) involving professionals who deliver care directly to a patient or a group of patients [1,2]. Technologies used include various types of ICT, like e-mail, messaging systems, video communication systems, smartphones, tablets, wireless monitors, and other forms of telecommunication technologies [3]. The main purposes comprise the provision of education (self-management support), exchange of information between health care providers (transfer of images and medical data), and facilitating contact with health professionals (continuous support from a distance) [4]. Telemedicine is said to facilitate access to relevant target groups and improve overall effectiveness of care [4,5]. As patients with chronic diseases like type 2 diabetes (T2D) or hypertension need continuous and individualized care, they are seen as the ideal target group for the use of digital health interventions like telemedicine [6,7].

However, evidence-based guidance on the effectiveness of telemedicine or specific functions (or components) in patients with diabetes or hypertension is still lacking [8,9,10]. Among others, heterogeneous patient populations, the diversity of telemedicine phenotypes in use and the settings they are used in hamper the evaluation of digital health interventions [11]. Additionally, a comparison of study results in order to provide evidence-based recommendations is further complicated by the diversity of applied study designs [12,13] and the heterogeneous definitions of “telemedicine” [1]. Consequently, a recent umbrella review conducted by the authors indicates that although telemedicine was found to potentially improve clinical outcomes in patients with diabetes or hypertension, the methodological quality of studies and overall certainty of subgroup-specific effectiveness were considered as moderate to low [14].

According to the evidence-based medicine manifesto, low-quality research may lead to poor information for evidence-based decision-making [15]. In a recent BMJ essay, authors state that the “continuing ethical, scientific, and economic deficiencies of medical research remain scandalous” [16]. Chalmers and colleagues identified heterogeneous sources of research waste, e.g., studies planned with inadequate statistical power and inconclusive results. At the same time, they argue that as long as the ideas for research are transparently developed, prioritized and warranted, the risk of research waste is limited [17]. Other experts argue that a synthesis of high-level evidence may help to inform policy and practice decisions [18]. One potential contribution to reduce the number and impact of potential research waste is adequate funding-decision-making. As such, Nasser and colleagues suggest different steps on how to define research priorities [19]. They propose (1) a pre-definition of stakeholders whose opinions and priorities should be taken into account, (2) the use of previous priority-setting exercises as a starting-point, (3) the use of available methodological reviews, (4) access to relevant data and qualification to conduct the priority-setting, (5) intervals to pilot, assess, revise, and update the priority-setting, and (6) sharing findings and experiences in lively discussions in the relevant community [19].

In order to inform policy and practice decisions, our approach combines qualitative analysis following a recent systematic analysis of the literature with further insights of experts in the field. Thereby, we aim to identify, categorize and prioritize specific implications for future research in telemedicine for diabetes and hypertension.

## 2. Materials and Methods 

We conducted a mixed method study [20] comprising three individual steps: (1) an updated literature search of an umbrella review on the effectiveness of telemedicine in three chronic conditions, (2) qualitative analysis of quotes on future research needs taken from the identified records, and (3) an online survey with practitioners and health care researchers designed to prioritize the identified research needs.

### 2.1. Umbrella Review Update of Relevant Literature

An update of a recently published umbrella review was conducted to identify relevant literature [14]. As such, systematic reviews or meta-analyses of RCTs evaluating the effectiveness of telemedicine in at least one of the target diseases (diabetes, hypertension and/or dyslipidemia) in adults were searched again in the databases Pubmed, Embase, and the Cochrane Library. The database search was complemented by a search of reference lists of the identified records and a hand search of major digital health journals (e.g., the Journal of Medical Internet Research). 

### 2.2. Qualitative Analysis—Data Extraction and Categorisation of Future Research Implications

Quotes dealing with future research implications were extracted from the discussion sections and conclusions of included records. To ensure topicality, only records published after 2015 were considered. Inductive formation of categories according to Mayring’s method of qualitative content analysis was applied [21]. The final codebook depicts those categories along with descriptions and examples per category (see annex, section I). Three researchers (PT, NM and LH) first familiarized themselves with the transcripts, and afterwards, independently identified relevant quotes. In a first iteration, underlying patterns and recurring schemes were identified and labeled, so that categories and subcategories emerged. After discussion and relabeling of these categories, examples taken from the extracted quotes were added. Coding and categorization were discussed among all three coders until agreement could be achieved. Reporting of results follows the Standards for Reporting Qualitative Research (SRQR) [22] and the consolidated criteria for reporting qualitative research (COREQ) [23].

### 2.3. Online-Survey—Prioritisation of Identified Categories

In a last step, the identified categories were used to develop an online-survey to prioritize research needs from the perspective of health care professionals, funding agents and researchers in the field of evidence-based health care. The questionnaire was subject to a cognitive pretest with five participants in January 2020 [24]. The survey was available in German and conducted online by using the soscisurvey platform. It was distributed via a questionnaire link among research, health care and health insurance facilities associated with the junior research group Care4Saxony. In addition, the Twitter channels of several German professional associations especially in the field of therapeutics and diabetes care were used to spread the link. The survey first ran from 31 January 2020 to 29 February 2020 and was then extended until 16 March 2020. 

For the dichotomous decision whether a need was important or not, cross-tabulations were computed, based on which Chi² statistics could be calculated in order to compare the priorities of different telemedicine experience levels. For the rating of the importance, the Likert scale from 1 to 6 was treated as a metric scale based on the almost equal variance (homoscedasticity) and the minimally interval character of the scale, so that t-tests could be applied [25]. 

A factor analysis to reduce the single research needs to overarching factors was applied. Factors were calculated from the 18 research needs considered important by the participants. In preparation, the bivariate variables for the research needs were first standardized to serve the demands of the factor analysis. Data analysis was done using SPSS 23 for windows. 

### 2.4. Mixing of Qualitative and Quantitative Data

Our mixed methods design is multi-phased in so far as the collection of quotes on further research needs is quantitative, whereas the in-depth analysis through content analysis is a qualitative process (explanatory data collection). The latter, in turn, informed the development of a standardized and quantitative questionnaire (exploratory data collection). Then again, the results of the questionnaire allow for a quantification of the research needs gleaned from the qualitative content analysis [20]. The presentation of the results leads to a Joint Display of the statistics-by-themes kind [26]. Mixing is not yet achieved in the purpose of the study, but during data collection, analysis and the drawing of conclusions [27].

## 3. Results

### 3.1. Umbrella Review Update of Relevant Literature

After updating the search and again applying eligibility criteria of the recently published umbrella review [14], 10 relevant new records [28,29,30,31,32,33,34,35,36,37] were identified. Consecutively, 35 records, namely systematic reviews (*n* = 9), meta-analyses (*n* = 5), systematic reviews and meta-analyses (*n* = 18) or other designs like meta-regressions (*n* = 2) or network-meta-analyses (*n* = 1), were included for further analysis. The included records focused on patients with diabetes (*n* = 24), hypertension (*n* = 6) or chronic diseases including at least one or both target diseases (*n* = 5).

### 3.2. Analysis of Future Research Implications

Of the 35 eligible records, 32 reported on future research implications. Their categorization in a qualitative content analysis revealed five categories as listed in Appendix A.

The first topic has five subcategories which further extrapolate the need for more rigorous research. Those are study designs, costs/cost-effectiveness, long-term effects, user satisfaction and adherence to reporting standards. The first subcategory “study designs” addresses non-reliable data on the effectiveness of telemedicine, calling mostly for more rigorously conducted studies. Within this subcategory, those quotes calling for larger studies (with a higher number of participants) [28] and quotes on both more rigorous RCTs [34,35,36,38,39,40] and more pragmatic observational studies are summarized [41]. The second subcategory describes the need for more data on direct and indirect costs of digital supported or delivered interventions. Essentially, the subcategory summarizes calls for a proper evaluation of the cost-effectiveness of telemedicine interventions [28,30,34,35,41,42,43,44] as well as their impact on health service utilization (e.g., hospitalization) [34,45]. Subcategory 3 states that there are limited data on the long-term effects of telemedicine and mobile health (mHealth) solutions, which should be gained by longer-lasting studies with an adequate length of follow up (e.g., >1 year) [29,36,41,46,47]. The fourth subcategory calls for an improved understanding of user acceptance of and satisfaction with telemedicine applications [38,39], which could increase engagement with intervention modalities [37]. Improved reporting of study details is described by authors whose reviews or meta-analyses have been subsumed in subcategory 5. This includes reporting data needed for effect size estimations [48], the intervention conduct [45], the uptake of the intervention by the participants [40] and the role of involved clinicians [40].

The second topic covers the mechanisms defining effective telemedicine interventions, the multiple features of these interventions and their combinations. Two subcategories were identified. The first subcategory calls for an improved understanding of the effectiveness of single components of the studied interventions. This refers to frequency, duration and delivery mode (e.g., via HCP, SMS, SNS) of feedback [34,36,41,49], as well as components of the intervention in general [37] and their combinations(s) [29,36,39,47] and contextual distinctions [29,37]. Some authors call for investigating strategies of tailoring [36], behavior change techniques the interventions should be based on [33], and the effectiveness assessment of specific elements like gamification and social media [41,50]. The second subcategory points to a need for a more holistic [29], standardized [37] and theory-grounded [31,36,50] assessment of interventions’ features and functionalities.

The third topic refers to an improved understanding of patients’ characteristics being associated with the overall effectiveness [33,51], i.e., their technology acceptance [28] as well as with initiating and sustaining the use of the intervention [30,39]. Some authors call for different [46] or more diverse patient subgroups [49], e.g., by performing studies in a larger variety of countries [29]. Others recommend improving the tailoring of interventions to individually address different subgroups, such as patients with high baseline HbA1c [36,42].

Ethical issues and safety were identified as the fourth topic. This includes threats to data protection [39] and the limited evaluation of interventions’ safety [30,41,42] (e.g., by considering adverse events [52]).

The fifth and last category underlines the need for translation of findings into real-life. Authors recommend more simple interventions [53,54], large-scale studies evaluating feasibility [55], as well as improved implementation, either via integration into existing technologies [55] or by providing reasonable timeframes for translation of findings into practice [56].

### 3.3. Online-Survey—Validation and Prioritisation of Identified Categories

#### 3.3.1. Descriptive Statistics of Study Sample

Overall, 188 participants took part in the survey, of which 86 (46%) finished it. Most participants quitting the survey did so after the second page, right before they were asked for previous experience with telemedicine applications. Appendix A shows a description of the study sample in terms of the occupational background and previous experience in the development, use and evaluation of telemedicine (Appendix A). Of the 86 cases included, almost two-thirds were constituted by ergo- (*n* = 24) and physical therapy (*n* = 24) or medical doctors (*n* = 8) (Professional occupation could be input freely and was coded for analysis) (see Appendix A). The majority (*n* = 62) subscribed to the definition of telemedicine provided by Sood and colleagues [1,2] (A minority either considered telemedicine to be used for data storing and sharing (*n* = 5), for self-monitoring/-tracking (*n* = 4) or for making medical decisions for diagnosis or therapy (*n* = 3). *n* = 5 subscribed to none of the definitions mentioned, while three stated they were not sure and four skipped the question entirely).

#### 3.3.2. Ranking of Future Research Needs

Appendix A depicts the ranking of the future research needs according to the participants of the survey. The exact wording of each question within the survey questionnaire, as well as explanations provided for non-common terms can be found in the Appendix A. Taking into account both the number of cases a need was chosen and the mean importance rating (using mentions by half of the sample, i.e., 41, as a threshold), the most pressing future research needs are data security, patient safety, patient satisfaction, implementation strategies and longer follow-ups.

Appendix A shows details on the prioritization of pragmatic study designs and study characteristics to be reported according to the participants. The most important study designs are waiting list control group designs (*n* = 16), where the intervention is distributed to the whole sample at different points in time to minimize drop-outs in the control group. The most important study characteristic to be reported is the data needed for effect size estimation (*n* = 24).

#### 3.3.3. Factor Analysis

A correlations matrix showed that several research needs were correlated significantly, e.g., the need for larger study samples and the reporting of relevant study characteristics (Spearman’s *r* = 0.413, *p* < 0.01). Consequently, the Kaiser–Meyer–Olkin criterion for sample adequacy is 0.69 (*p* according to the Bartlett-test for homoscedasticity < 0.001) and therefore above the established minimum of 0.6 [57]. 

Varimax rotation was used to extract orthogonal vectors, i.e., disjunctive factors. Eigenvalues and the cumulative variance explained suggest six underlying factors. The cumulative variance of the research needs explained by the presence of the six factors is 60.81. The six factors are depicted in Appendix A, along with the factor loadings of the variables they include. The fact that the factor “data security” only covers one variable shows that this research need has little to do with the other ones and is a rather singular issue.

#### 3.3.4. Differences in Prioritization According to Experience with Telemedicine

Having had any experience with telemedicine previously (as depicted in Appendix A) is correlated with considering patient safety a future research need (Pearson’s Chi² = 3.77, *p* = 0.05), yet has no impact on any other research need being mentioned. However, there are significant differences in the degree of importance for some of the research needs between those with previous telemedicine experience and those without. Those with previous telemedicine experience find research on data security (*t* = −2.28, *p* < 0.05), implementation strategies (*t* = −2.79, *p* < 0.01), interoperability (*t* = −1.89, *p* < 0.01) and patient satisfaction with a given telemedicine application (*t* = −2.12, *p* < 0.05) more important than those without. A more detailed assessment according to which kind of previous experience with telemedicine was present yielded no significant differences whatsoever.

### 3.4. Joint Display

Read from left to right, the joint display (Table 1) first covers the categories and subcategories developed inductively by qualitative content analysis. The subcategories are further described by quotes on future research needs taken from the 32 included records. After that, the survey results are depicted in terms of the rank a research need achieved, its mean importance, and the variance of each mean. Finally, the factor each need belongs to is depicted. 

## 4. Discussion

After updating the umbrella review and applying the eligibility criteria, quotes on future research implications were extracted from 32 systematic reviews and meta-analyses published after 2015. The results yielded five categories, as well as subcategories, covering a need for high quality studies, comprehensive technology assessments, in-depth considerations of patients’ characteristics, ethics and safety as well as implementation strategies. The online survey revealed that data security, patient safety, patient satisfaction, implementation strategies and longer follow-ups were seen as the most pressing future research needs. The needs and especially the degree to which they were deemed important differ between respondents with and without telemedicine experience, which suggests that having been involved in telemedicine projects before leads to a better knowledge of potential pitfalls [58].

The identified future research implications calling for high quality effectiveness studies are similar to those systematically assessed by the group of Ekeland and colleagues. They also call for larger studies as well as mixed methodological approaches to assess complex telemedicine interventions [12]. Our results show that many study groups criticize the presence of small pilot studies and limited data on mid- to long-term outcomes. This was also a result of the assessment of publication bias in the included studies analyzed in the previous umbrella review [14], highlighting a dominance of smaller studies with larger effect sizes. Some of the included records also criticize inadequate reporting of study data, which is also attributed to an inadequacy of existing reporting guidelines [40,45,48]. This criticism, e.g., on the applicability of CONSORT criteria for eHealth trials, has also been articulated before [59]. This has led to new suggestions for the assessment of telemedicine applications (e.g., MAST [60]) as well as an updated CONSORT-EHEALTH checklist for authors and editors to improve reporting of RCTs on web-based and mobile interventions [61]. 

The recent consensus report of the American Diabetes Association (ADA) and the European Association for the Study of Diabetes (EASD) acknowledges the emerging factor of technology but simultaneously calls for research on the mechanisms of enhanced monitoring of blood glucose and other variables to individually adapt treatment [62]. Our results indicate conflicting perspectives pointing either to the need for more robust evidence (e.g., in terms of more rigorous RCTs) or more pragmatic, engaging and scalable intervention designs favoring rapid cycle design evaluations [13]. Potential options to overcome this dilemma are the combination of RCTs with qualitative assessments [63], or the application of adaptive study designs in telemedicine research [64,65]. On top of that, other scientists call for an adaptation of methods like meta-analyses to the requirements of diabetes technology [66]. This somewhat discordant state of the art is mirrored by the results of our survey where adequate tailoring is ranked high (2nd), yet study designs, be they rigorous or more pragmatic, rank low.

Although telemedicine is expected to close gaps in health care delivery, which arise e.g., due to demographic change and regional shortage of services (especially in rural areas) [67,68], the problem often referred to as pilotitis/pilotism is prominent. This refers to the presence of many small and/or regional projects proving to be beneficial for a certain study population, with limited large-scale implementation [69,70,71]. Fittingly, both ADA and EASD state that the perspective of implementation science in the prevention and care is lacking behind [62], a perspective shared by our survey participants who prioritize research on implementation strategies highest. Records included in our analysis suggest to develop more simple interventions [53,54], carry out large scale studies evaluating feasibility [55], or integrate new interventions into existing technologies [55]. A recent review of telemedicine for the self-management of hypertension supports this by concluding that lack of evidence, difficulties to maintain self-management over time, and lacking long-term results were common barriers [72]. The respondents of our online survey prioritized similar needs, especially those concerning implementation strategies and the related needs for assessments of patient safety and satisfaction as well as data security. These somewhat patient-centered needs correspond well with the future research agenda for personalized telehealth proposed by Dinesen et al. [13].

Our results support and emphasize the recommended systematic and structured guidelines for digital health apps as well as an improved evidence base for digital health interventions by ADA and EASD [73]. Additionally, our results show that considering Patient-Reported Outcomes, supporting implementation science and continuously involving end-users in the development process of digital health interventions (user-centered design) are seen as relevant strategies, which is in line with the evidence tiers of the recently published NICE framework for digital health evaluation [74]. The results of our online survey further indicate that the priority of certain research needs may depend on the level of experience in telemedicine use, evaluation and development. Therefore, it is advisable to include different stakeholders with different levels of experiences in panels to define a necessary research agenda for telemedicine in diabetes and hypertension. The procedure for developing Core Outcome Sets can be used as a blueprint [75]. Especially the six factors of research needs derived from the survey not only support the results of the qualitative content analysis but can also serve as fields of action for future policy agendas in telemedicine implementation and evaluation. As such, our work may be used as a starting point for the systematic and structured consensus building by representative panels. Given the fact that all the research needs gained high values for importance according to our survey participants, the matter of consensus building is pressing.

Despite the strengths of the applied methods and their joint analysis, our work has limitations. As described in the recent umbrella review, the initial search and inclusion process can be considered as a limitation. Our approach focused on high quality records (systematic reviews and meta-analyses). As our analysis heavily relies on the conclusions and recommendations from the included records, we may have unintentionally adopted biased inferences. However, as we analyzed the included records and their primary studies, this risk can be considered as low. Additional limitations arise from the conducted qualitative content analysis, which may be influenced by the individual backgrounds and unintended expectations of the authors during the inductive coding process. Although this risk of bias is inherent to qualitative research, it was mitigated by three coders categorizing the quotes on future research needs. We used an online survey to prioritize the identified research needs from different perspectives. The link to the survey was sent out to potential participants and also included larger networks, federations and associations to maximize the number of potential participants. However, we may have missed relevant perspectives, because people were not available, were not reached by our snowball sampling approach or generally do not respond to digital surveys. Additionally, the online survey was only available in German, limiting the generalizability of our findings. The snowball sampling approach itself does not allow for drawing inferences to a certain population, as data are not representative.

## 5. Conclusions

A mixed-method approach was applied to identify, categorize and prioritize future research implications based on systematic reviews and meta-analyses published after 2015. According to the qualitative analysis, authors call for high-quality studies, comprehensive technology assessments, in-depth considerations of patients’ characteristics, ethics and safety as well as implementation strategies. Both the categories form the qualitative content analysis and the overarching factors gleaned from the survey results can serve as key topics of a future telemedicine research agenda.

## Figures and Tables

**Table 1 nutrients-12-01340-t001:** Joint display of qualitative and quantitative results.

Topic	Subcategory	Content	Examples	Survey Results		Factor
				Rank	Mean Importance (Variance)	
Need for high quality studies including specific outcome measures	Study Designs	More rigorous RCTs [34,35,36,38,39,40];	“This indicates that future studies should consider some essential criteria, including a sufficient number of participants and duration time, concealment and randomization procedures, blinding of the assessor, and low attrition rates [38].”	9	5.21 (0.96)	I
		Studies with higher number of participants [28]		15	5.08 (0.95)	
		Pragmatic Study Designs [41]	“The long-term effects (>1 year) of diabetes apps are still unknown and need to be investigated in more pragmatic observational studies [41].”	13	5.04 (0.92)	
	Long-term effects	Longer follow-ups [29,36,41,46,47]	“Future well-designed intervention studies with adequate length of follow-up are required to assess these important endpoints [45].”	5	5.05 (0.95)	
	Costs/cost-effectiveness	Analyses of cost-effectiveness [28,30,34,35,41,42,43,44]	“In future studies, helping decision makers prepare well-informed reimbursement decisions by analyzing cost-effectiveness and safety is recommended [30].”	11	4.97 (1.07)	II
		Analyses of changes in health service use [34,45].	“Because of a lack of studies, conclusions cannot be drawn about effects on mortality and hospitalizations [45].”	7	4.64 (1.87)	
	Adherence to reporting standards	Adherence to reporting standards [45]	“The application of the TIDieR checklist highlights a need for better reporting of telehealth interventions, because many trials did not report important logistical data relating to intervention conduct [45]”.	12	4.81 (1.62)	
		Effect size estimations [48]	“We suggest to authors of future intervention studies, particularly with baseline imbalance, to report detailed information that allows authors of systematic reviews to calculate ANCOVA effect size estimates or, ideally, to provide access to IPD [48].”	-^1^	n mentioned ^1^/n total	
					24/27	
		Intervention uptake [40]	“Future studies should report specific details relating to uptake and engagement of the participant with the intervention, the application (including process of nutrition data entry and use of a database) and the involvement and role of clinicians to enable reproducibility and comparison with other applications and studies [40].”	-^1^	18/27	
		Role of involved clinicians [40]		-^1^	22/27	
	User satisfaction and technology acceptance	User acceptance [38,39]	“It is important also to assess and understand users’ satisfaction with and acceptance of these apps [38].”	10	4.65 (1.24)	
		Patient satisfaction [38,39]		4	5.3 (1.14)	V
Need for comprehensive technology assessment	Understanding the prerequisites, mechanisms and combinations	Analyses of intervention components [34,36,41,49]	“Future studies assessing the effectiveness of apps should focus on apps that incorporate more comprehensive functionalities, that are identified in this review as the most promising functionalities for self-management of hypertension, including self-monitoring, reminders and alerts with either automatic feedback or educational information or both [38].”	16	4.81 (1.9)	
		Adequate Tailoring [36,42].	“This study has significant implications for future research. Investigating the effects of different tailoring strategies for diabetes self-management is important and future research should further explore the relationships between tailoring strategies and other intervention components [36].”	2	5.11 (1.83)	
	Comprehensive assessment of features and functionalities	Basic Theories and/or frameworks [29,31,36,37,50]	“Future research should clearly identify and report the explanatory frameworks, mechanisms and theories for the social network interventions being tested [50].”	12 ^2^	n mentioned/n total	
					17/27	
Need for in-depth considerations of patients’ characteristics and more diverse study populations	-	Improved understanding of patient characteristics [28,33,39,51]	“Future studies also need to examine whether certain patient characteristics are more likely to result in initiating and sustaining the use of diabetes apps [39].”	8	4.97 (1.68)	II
		More diverse study populations [29]	“performing studies in other countries and places is suggested [29]”	18	4.5 (1.61)	III
Ethics and Safety	-	Data security [39]	“Ethical considerations for the risk of data privacy also need to be carefully addressed [39].”	3	5.47 (0.94)	VI
		Patient safety [30,41,42,52]	“Safety issues such as hypoglycemia and other adverse events are being overlooked and need attention in future investigations [52].”	6	5.25 (1.27)	II
		Implementation strategies [56]	“Further research should be conducted to provide more valid evidence for the effects and sustainable implementation of telehealth [42].”	1	4.76 (1.3)	IV
		Evaluation of implementation strategies [42,55]		14	4.68 (1.56)	
		Interoperability [55]	“(…) future studies need to integrate diabetes-related functions to existing technologies, such as developing diabetic apps, which could be directly installed into patients’ own mobile phones, rather than developing new types of phones [55].”	17	4.45 (3.07)	III

The joint display covers the categories, subcategories, their description and example quotes taken from the 32 included records. Ranks show the results of the survey. The following six factors (right column) were found: I = Evaluation of effectiveness; II = Diversity of outcomes and standardized reporting; III = Research Planning; IV = Implementation science; V = User-centered design; VI = Data security; ^1^ Specific reporting standards were asked for in a separate question only applicable to those who found them important. 2 The item was part of the special question on reporting standards only applicable to those respondents who found them useful.

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
