# Peer review of "Research Implications for Future Telemedicine Studies and Innovations in Diabetes and Hypertension—A Mixed Methods Study"

_nutrients, 2020, doi:10.3390/nu12051340_

Round 1

Reviewer 1 Report

This is a very interesting paper as it offers an unbiased view of the current level and the limitations of studies regarding telemedicine applications for diabetes and hypertension.

The authors should be congratulated for their findings and their critical evaluation in order to see real progress in this field in the future.

Author Response

Dear Reviewer 1,

Thank you very much for your positive feedback. We are happy that our work met your high standards and do appreciate your congratulations.

All the best

Reviewer 2 Report

All topics are well discussed. However there are some points to clarify and comment.

LINE 106

Choosing the German language for the survey in a on line survey may have limited the power of the research, considering that this survey has been sponsored with instruments like twitter, that has a worldwide coverage.

LINE 109

Which “professional associations” has been chosen?

LINE 152

“mHealth” is the first time that appears in the manuscript, what does it means and what does it comprehends for the authors? (I think mobile health?)

LINE 200

“According the participants” transform in “According to the participants”

LINE 312-318

In this discussion field it could be useful to discuss the limit of the survey submitted in a language that it’s not spoken in the worldwide scientific community. An English survey could reach more participants in such a subject like telemedicine, improving the research power. Choosing German language for the survey could limits the experience to a local reality.

LINE 349

This table show that the participants has low experience in telemedicine, this data should be considered more in the discussion considering the aim of this paper.

LINE 394-395

Font error for this citation

English should be simplified.

Author Response

Dear Reviewer 2,

Thank you very much for your review and the in-depth analysis of our manuscript. Below, we would like to address the single points raised:

LINE 106: Choosing the German language for the survey in a on line survey may have limited the power of the research, considering that this survey has been sponsored with instruments like twitter, that has a worldwide coverage.

RESPONSE: Thank you for this reflection. The German online survey is certainly limited in its scope and reach. However, due to diverging health system structures and different levels of telemedicine implementation, the focus on Germany supports the internal validity of our approach. We added a comment on the limited reach of the survey due to the language chosen in the limitations section. In a potential next research step, a multi-lingual online survey may be used.

LINE 109: Which “professional associations” has been chosen?

RESPONSE: As participants of our survey were guaranteed privacy and anonymity of their data and this might be compromised by naming the associations we contacted in detail, we would like to refrain from doing so now. Thank you for your understanding.

LINE 152: “mHealth” is the first time that appears in the manuscript, what does it means and what does it comprehends for the authors? (I think mobile health?)

RESPONSE: mHealth refers to mobile health. We added this in the text.

LINE 200: “According the participants” transform in “According to the participants”

RESPONSE: Thank you. We inserted the missing “to”.

LINE 312-318: In this discussion field it could be useful to discuss the limit of the survey submitted in a language that it’s not spoken in the worldwide scientific community. An English survey could reach more participants in such a subject like telemedicine, improving the research power. Choosing German language for the survey could limits the experience to a local reality.

RESPONSE: Please consider our response to your first comment and find our added reflection in the limitations section. Thank you.

LINE 349: This table show that the participants has low experience in telemedicine, this data should be considered more in the discussion considering the aim of this paper.

RESPONSE: Thank you for this important remark. Our overall aim was to identify, categorise and prioritise specific implications for future research in telemedicine for diabetes and hypertension. In our discussion, we highlight the important finding that the needs and especially the degree to which they were deemed important differed between respondents with and without telemedicine experience. We also recommend including different stakeholders with different levels of experiences in panels to define a necessary research agenda for telemedicine in diabetes and hypertension. From this perspective, we are confident that we already stressed this issue and its implication in our original version of the discussion.

LINE 394-395: Font error for this citation

RESPONSE: Thank you. We adapted the font size for reference 3.

All the best

Reviewer 3 Report

This is an excellent paper that is very timely given the pressing need for increased use of telemedicine in the era of Covid 19. I have no hesitation in recommending prompt publication of this paper

Author Response

Dear Reviewer 3,

Thank you very much for your positive feedback. We feel honored that you recommend our work for prompt publication. Indeed, telemedicine has a strong potential to support patients with chronic diseases like diabetes and hypertension (which are at the same time at elevated risk for a rather sever course of a Covid 19 infection). As such, our identified and prioritized research needs may certainly guide relevant next steps in this perspective as well.

Thank you and best wishes.

All the best